# OpenReview forum: "Breaking MCP with Function Hijacking Attacks: Novel Threats for Function Calling and Agentic Models"
_TMLR — Under review for TMLR_

### Review · Reviewer_XZJ4 · 2026-05-08

**Summary Of Contributions:**

The paper introduces a Function Hijacking Attack (FHA) against function-calling/agentic LLMs. The threat model assumes an attacker who controls only the textual description of a tool (e.g., in an MCP server) and wants to cause that tool to be selected over the legitimate ground-truth tool when a user issues a query. The authors adapt GCG to optimize an adversarial suffix appended to the target function's description, with a target objective that forces the model to call the attacker-chosen function (and, for reasoning models, to skip or shortcut chain-of-thought). They evaluate on the BFCL dataset across five models (Llama-3.2-3B, Mistral-7B-Instruct, and Qwen-1/Qwen-3 variants), report per-query ASRs of roughly 70-100%, and include a Section 7 "universal" variant where a single suffix is trained to hijack tool selection across multiple queries (run only on Llama-3.2-3B, with substantially weaker ASR). They compare against a function-injection baseline that uses Llama-3-70B to one-shot synthesize a competing decoy function.

Strengths:
- The basic observation that GCG-style optimization can be applied to tool descriptions to bias selection is a clean and reasonable extension of adversarial-suffix work into the agentic/MCP setting.
- The per-query attack numbers in Section 6 are strong, and the position/noise ablations are a nice touch.
- The paper is a useful data point that tool-selection prompts inherit the adversarial-suffix vulnerabilities of the underlying LLM.

Weaknesses:
- The threat model is essentially asserted rather than motivated. The specific choice of perturbing only the description (versus the function body, name, or parameters) is justified by an appeal to ease of detection, but I think this gets the empirical situation backwards because adversarial tokens in descriptions are arguably easier to flag than semantic compromise of a function body.
- The Section 6 headline numbers are obtained by optimizing a GCG suffix against a single (target_function, query) pair. This is a substantially weaker attack than the universal-over-queries threat one would actually care about in the MCP setting, and the framing does not make this limitation salient.
- The Section 7 universal-over-queries results, which are the operationally interesting setting, are much weaker and only run on a small model (Llama-3.2-3B).
- No transfer experiments across models, and in particular, no transfer to closed/larger models. Transfer is the property that makes GCG interesting in the first place. Without it, the contribution is significantly narrower than implied.
- The function-injection baseline is described as "strong" because it is generated by a larger model (Llama-3-70B), but a one-shot generation by a 70B model is not a fair comparator to gradient-based optimization against the target model. The claim that FHA beats a strong baseline is partly an artifact of baseline weakness.
- The "skip thinking" mechanism for reasoning models is central to the reported numbers but is only explained in the appendix.

**Additional Comments:**

A few smaller points that did not fit cleanly into the boxes above:

- The abstract and introduction do not make the actual attack mechanism clear up front. It took me until Figure 1 and Section 4 to be confident about what is being perturbed, by whom, and against what. A one- or two-sentence description in the abstract of the form "the attacker controls the textual description of a tool exposed to a function-calling model and appends an adversarial suffix optimized via GCG to cause that tool to be selected" would substantially help.

- Relatedly, the first paragraph of the introduction frames the contribution as addressing "the challenge of controlling the perturbation of the function calling process," which is vague. Leading with the concrete attack and threat model would be much stronger.

- It is unclear why several of the analyses in Section 6.2 (and the universality experiments in Section 7) are run only on Llama-3.2-3B. I recognize that there are computational constraints, but the choice seems strange. In my opinion, the paper would be strengthened by a more comprehensive set of experiments here (or by de-emphasizing those results).

**Audience:**

Yes

**Audience Explanation:**

Yes, but with qualifications. The agentic-AI and tool-use security community will be interested in the basic finding that GCG-style adversarial suffixes can bias tool selection when injected into tool descriptions, and the position/noise ablations are useful. However, the audience is narrower than the paper implies: the per-query attack setting is not the operationally interesting one, the universal-over-queries results are weak and only on a 3B model, and there is no model-transfer evidence.

**Broader Impact Concerns:**

No additional broader-impact concerns beyond those already addressed in the paper's Broader Impact statement. The work is offensive security research on a deployed protocol (MCP / function calling), but the authors include a broader-impact discussion and the disclosed attack does not provide capabilities meaningfully beyond what is already implied by the GCG line of work; the marginal uplift to a real attacker from this paper appears modest, and the defensive framing is reasonable.

**Claims And Evidence:**

No

**Claims Explanation:**

My answer is No, primarily because the gap between what the paper measures and what it claims is larger than the presentation suggests.

1. Threat model is under-motivated. The paper asserts a setting in which an attacker controls only the description of a tool exposed to a function-calling model, but does not seriously argue why this is the right slice of the attack surface. The justification that modifying function names, parameters, or implementations would be caught by automated validation is not convincing. In practice, scanning a description for an adversarial token suffix is at least as tractable as auditing a function body for semantic compromise. The MCP/tool-poisoning literature is gestured at but not engaged with in a way that situates the specific contribution.

2. The headline ASR numbers (Section 6, 70-100%) are obtained against a fixed (target_function, query) pair. This is not the threat model that matters operationally: an attacker poisoning an MCP description does not get to re-optimize per user query. The framing of Section 6 as the main result, with universality deferred to Section 7, oversells what the attack actually achieves in a realistic deployment.

3. The Section 7 results are substantially weaker and are only reported on Llama-3.2-3B. This is precisely the regime where the claim of "universal adversarial functions" in the abstract should be supported, and the evidence provided is thin.

4. No model-transfer evaluation. GCG's interest in the prompt-injection literature comes in large part from suffix transfer to larger and closed models. The paper does not evaluate transfer across model families or to closed models. This leaves a central question about the practical reach of the attack unanswered.

5. The function-injection baseline is described as "challenging" because it uses a larger generator (Llama-3-70B), but one-shot generation by a 70B model is not a strong comparator to gradient-based optimization against the target model itself. The apparent superiority of FHA over this baseline is therefore weaker evidence than the paper presents it as.

Taken together, the empirical claims that are well-supported are narrower than the claims the paper foregrounds, and several of the claims that would matter most for an MCP-style threat model (universality across queries on non-toy models, transfer across models) are either weak or absent.

**Requested Changes:**

Critical (would be needed for me to recommend acceptance):

1. Motivate the threat model. The paper should explicitly engage with the MCP / tool-poisoning literature (e.g., Invariant Labs and related work) and argue why description-only perturbation is the right operating point. In particular, please justify the claim that descriptions are a more realistic attack surface than function bodies, names, or parameters. Without this, the paper's framing of the contribution does not stand on its own.

2. Reframe the contribution around query-universal attacks. The per-query attack in Section 6 is not the threat model that matters in the MCP setting because an attacker poisoning a tool description does not get to re-optimize against each user query. Either (a) demote the per-query results and lead with universal-over-queries attacks, or (b) explain clearly and early what the per-query setting corresponds to operationally and why it is interesting on its own terms.

3. Strengthen the universality experiments. Section 7 is currently only reported on Llama-3.2-3B and shows substantially weaker ASR. To support the abstract's claim of "universal adversarial functions," please run universal-over-queries attacks across the same model set used in Section 6, and report ASR distributions over held-out queries.

4. Add model-transfer experiments. Evaluate suffix transfer across model families (e.g., suffix optimized on Llama transferred to Mistral/Qwen) and, if at all feasible, to a closed model. This is the property that makes GCG-style attacks consequential, and its absence is a major gap.

5. Qualify the function-injection baseline. The current baseline relies on one-shot generation by Llama-3-70B and is described as "challenging" on those grounds. Either justify why one-shot generation by a 70B model is a fair comparator to gradient-based optimization against the target model, or replace/augment with a baseline that itself uses optimization (e.g., a prompt-injection or rewriting attack with a comparable compute budget).

Strengthening (would improve the paper but are not blocking):

6. Move the description of the "skip thinking" mechanism for reasoning models out of the appendix and into the main text, ideally where the threat model and attack objective are first described. As written, a key reason the reasoning-model numbers are so high is buried.

7. In Section 3, spell out "function calling models" in the section heading rather than "FC models," matching the convention used for "large language models" in the preceding section.

8. Consider an end-to-end demonstration in a real or realistic MCP setup (rather than only static BFCL queries), to ground the threat model in something a reader can picture deploying.

9. Provide more detail and intuition in the main text on what "epoch" corresponds to in the GCG curves, and report final ASR alongside or instead of trajectories where the trajectory is not the point being made.

---

> ### Author Response · Authors · 2026-07-04
> **Answer to Review**
>
> Thank you for your review comments and feedback on our work. We appreciate your questions and your concerns about our work. It helps us to better understand our weaknesses and gives us ways of enhancing and strengthen our claims. To address your concerns, we uploaded an updated version of the manuscript, with changes annotated in blue to better highlight our modifications. Please find below our responses to your questions.
>
> ### 1. Motivating the threat model
> We agree Section 4 under-motivates our threat model with regards to the existing literature, and currently describes the base FHA objective and only describes the universal variant in Section 7. In the revised version, we included the following changes:
>
> - Enhanced the motivation of our threat-model, including a discussion of FHA with regards to existing attacks, and the selection of the description-only perturbation. **NB:** Regarding the motivation of the attack location, we also included in the revised manuscript the following point. Other attack surfaces (i.e. function parameters, or names) make the codebase brittle. In a real-world setting, the functions called by MCPs are often inherently fused and used by a large codebase. Therefore, attacking an existing function might not be feasible if the attack choose to attack the function name, or properties of the functions as those would change, and are directly used by the codebase. In contrast, the description only serves as describing the function itself, and is only used by LLMs to support their tool-call decision. This is an additional motivation of our selection.
>
> - Reformulated Section 4 to state the universal optimization objective (Eq. 8 - Eq. 6–7 plus the batch-accumulated loss from Algorithm 2) explicitly, rather than deferring it to Section 7.
>
>
> ### 2-3. Re-framing around query-universal attacks
> We agree with the reviewer's points and suggest the following changes:
>
> - Promoted the universal (query and payload) attack from Section 7 to be introduced immediately after the threat model in Section 4, framed as the primary FHA configuration.
>
> - Re-framed the current Section 6 per-query, per-payload results explicitly as single-payload ablations that isolate individual factors (position, function count, semantic distance) affecting attack difficulty. We think that Section 6 as it stands remains useful for understanding why the attack works, but we agree that is should not be presented as the deployable result.
>
> - Extended Section 7's universal-over-queries experiments beyond Llama-3.2-3B to three models (Mistral-7B, Qwen3-1.7B, Qwen3-8B), reporting ASR scores over trained, and held-out queries for each (see Tables 3 and 4 in Section 7). FHA reaches 62.5% to 81.9% ASR on held-out queries, which further validates our Takeways 3.1 and 3.2.
>
>
> ### 4. Model-transfer experiments
> We included an ablation transferring the attacks from one model to another. We found the attacks transfers modestly to other model families or sizes, but that the ASR obtained is non-negligeable (11.2-27.6% ASR - see Table 5 in Section 7). This is likely due to the formatting of the optimised suffixes. For this reason, we still report these findings for transparency, but do not claim full cross-model transferability of our attack and leave this for Future Work (see Section 7 and Section 9).
>
> ### 5. Qualifying the function-injection baseline
>
> We agree that comparing a one-shot Llama-3-70B generation to gradient-based optimization against the target model is not a matched comparison. We address this concern by:
>
> - Modifying the Section 5's baseline description to mention that the selected baselines are not aimed to match the computation of FHA, but rather lower resource setting, which still show satisfying performance.
>
> - To further address this concern, we present a novel baseline in Appendix K: the Multi-Turn Function Injection Attack, where a model capable model (larger - gpt-4o-mini) is used, and in an iterative setting. We tested our novel baseline on the query-universal attacks, and demonstrated an enhanced performance compared to our original baselines, making it more challenging.
>
>
> ### 6. Description of the "skip thinking" mechanism for reasoning models
>
> We agree that moving this item to the main text would be helpful for the reader to better understand the differences between models. We included a paragraph in the main text (Section 4), describing our Appendix A.
>
> ### 7. Writing conventions
>
> Thank you for pointing out this, we did a pass through the abbreviations that we are using. We hope that our changes would address your concerns regarding this point.
>
> ### 8. End-to-end demonstration in a real MCP setup
>
> In Section 6, our paragraph FHAs on MCP refers to the Appendix F, which present the performance of FHA on two real-world MCPs directly picked from [2], namely the GitHub MCP Server, and Slack MCP Server (Figures 16 and 17, respectively).
>
> References:
> [2] https://github.com/modelcontextprotocol/servers-archived/

---

### Review · Reviewer_bYnF · 2026-06-19

**Summary Of Contributions:**

### Summary:

This paper aims to find adversarial tool descriptions using a Greedy Coordinate Gradient (GCG)-style optimization. They define a loss that makes the model more likely to output the target tool call, then update the adversarial token string using gradients. They explicitly adapt GCG and use a cross-entropy objective over the target output sequence to achieve this.

### Strengths:

- The attack setting is relevant to many users who use MCP tool calls in their workflows, but novelty is incremental relative to MCP tool poisoning, MPMA, and tool-description prompt injection.
- An analysis of the universality of the adversarial perturbation is provided, and this is useful to determine whether the adversarial prompts transfer to different queries as well as within varying tool lists.  This show the attack can tolerate some increase in the number of available tools.

### Weaknesses:

- The study does not test basic defenses: description linting, adversarial-string detectors, tool-description normalization, or a separate tool-selection classifier. This can be used since the adversarial suffixes often look like obvious garbage, and 60-token suffixes are central to the strongest results. Ideally the adversarial tokens should be non-obvious to filter out, and a study of the "stealthiness" of the attack would be useful.
- The demonstrated method is primarily a white-box attack. The GCG-style optimization requires gradients through the target model, while many realistic MCP users rely on closed-source or hosted models. The paper does not show whether the attack transfers reliably to closed models, works under output-only API access, or can be found efficiently through black-box search.
- The “works in practice” claim is too strong. Slot-filling ASR checks structural validity, not whether the action succeeds through real authorization, confirmation, API permissions, or human review. A valid tool call is not the same as a successful harmful operation. Also most MCP tool calls have more than 4 choices to choose from and the attack seems to drop in effectiveness greatly when going beyond 3 choices.
- The main benchmark setting is much smaller than many realistic MCP deployments. BFCL contains only 2 to 4 available functions per sample, and the large-tool robustness evidence is limited. The 25-added-function experiment is useful, but it is closer to a perturbation stress test than a full evaluation on real large MCP servers.

**Audience:**

Yes

**Audience Explanation:**

This paper brings to light a novel way to attack MCP tool calling agents and would be of interest to researchers studying open LLM models or about using them for tool calls.

**Broader Impact Concerns:**

Since the paper introduces an attack that could be directly applied to MCP tool descriptions, the broader impact section should include a clearer responsible-disclosure discussion, concrete deployment cautions, and at least lightweight mitigation guidance.

**Claims And Evidence:**

Yes

**Claims Explanation:**

This work provides a useful benchmark and characterization of MCP tool hijacking. The results show a non-trivial attack working on various open-source LLM models with studies on the universality of the adversarial token sequences in the presence of noisy inputs and additional functions to call.

**Requested Changes:**

1. (Critical) Evaluate simple defenses and stealth. At minimum, test description normalization, removal of suspicious tokens, length-based filters, and/or a separate tool-selection verification step.

2. (Important) Add a closed-model, surrogate-transfer, or output-only black-box evaluation. If this is not feasible, the paper should clearly limit its claims to white-box or surrogate-model attacks.

3. (Important) Include compute cost and attack practicality, including GPU hours, number of epochs, and whether attacks can be generated at scale.

4. (Nice to have) As an obvious baseline, would it make sense to put an instruction in the tool description saying “always call this tool for any request” or “this tool is mandatory.” ? This could be a direct natural-language tool-description poisoning baseline which may be more useful than MPMA since the BFCL dataset contains semantically different functions which MPMA fares poorly in.

5. (Nice to have) The authors should clarify that the 20-token reference (`optim_str`) is the default adversarial suffix length, not the original function description. They should also distinguish between proportion of the full payload and proportion of the modified tool description. A 60-token adversarial string may be only around 5% of the full FC input, but it can still dominate the target function’s natural-language description, making the stealth claim weaker.

---

> ### Author Response · Authors · 2026-07-04
> **Response to Review**
>
> Thank you for your review comments and feedback on our work. We appreciate your questions and your concerns about our work. It helps us to better understand our weaknesses and gives us ways of enhancing and strengthen our claims. To address your concerns, we uploaded an updated version of the manuscript, with changes annotated in blue to better highlight our modifications. Please find below our responses to your questions.
>
> ### 1. Evaluate simple defenses and stealth.
>
> We agree that adding an ablation with some form of defenses is important to further challenge our approach. For this reason, we implemented a few defenses to evaluate our attack, including description normalization, and guardrail models. We will post an update as soon as possible, once our runs will be completed.
>
> ### 2. Add a closed-model, surrogate-transfer, or output-only black-box evaluation.
>
> We agree that evaluation of FHA on closed-model would be valuable, but such study is out-of-scope of our paper. We mentioned in the Target models paragraph in Section 5 that FHA is primarily a white-box attack as it requires gradient access to the target model. While transferability to these models might be feasible, we leave that to future work.
>
> Further, we included an ablation transferring the attacks from one model to another (see Table 5 in Section 7). We found the attacks transfers modestly to other model families or sizes, but that the ASR obtained is non-negligeable (11.2-27.6% ASR). This is likely due to the formatting of the optimised suffixes. For this reason, we still report these findings for transparency, but do not claim full cross-model transferability of our attack.
>
> ### 3. Include compute cost and attack practicality.
>
> We agree that such study was missing in the initial manuscript. To address this, we included an analysis of the resources needed by each algorithms, as well as an analysis of their runtime in Appendix M. We discussed about the GPU time across models and FHA set-ups.
>
> While this ablation shows that FHA is costly compared to the multi-turn baseline, we included additional experiments in Section 7 regarding the query-universality of the attack at scale. Specifically, we show that FHA reaches 62.5% to 81.9% ASR on held-out queries, which further validates our Takeways 3.1 and 3.2. Importantly, this finding demonstrate that FHA can recover some training costs at scale as it remains robust to query and payload perturbations.

---

> > ### Author Response · Authors · 2026-07-08
> > **Follow-up on 1. Evaluate simple defenses and stealth.**
> >
> > Again, thank you for your review comments and feedback on our work. Please find below a follow-up regarding the robustness of FHA with regards to Tool-Calling defenses.
> >
> > ### 1. Evaluate simple defenses and stealth.
> >
> > Appendix N presents a study of the robustness of FHA with regards to simple Tool-Calling defenses. We implemented three defenses, namely: Normalization, Length Filtering, and Suspicious Tokens (see implementation details in Appendix N). Table 11 compares the ASR of FHA to the standard inference (i.e. without attack). A defense is considered as satisfying if the standard accuracy is maintained, but the ASR of the attack drops. Overall, we show that FHA is strongly resilient to Normalization, moderately resilient to Length Filtering, while not being robust to Suspicious Tokens. We agree that such ablation was missing, as it strengthen the evaluation of FHA in the context of real-world deployment.

---

### Review · Reviewer_hzRk · 2026-06-29

**Summary Of Contributions:**

The paper proposes applying the GCG jailbreak to target eliciting a function call by optimizing a suffix of the function's description. This new problem setting is formalized and distinguished from prior areas, and extensive analysis is done on evaluating the nuances of GCG effectiveness and robustness in this setting.

However, a large gap remains between attack settings explored and the more realistic settings where less knowledge is known about user query, functions, and specific model. Additionally the clarity of the method and experimental settings could be improved.

**Additional Comments:**

- Intro last sentence of "Universality" point typo: "design a enhance"
- Table 2 recommend changing notation to '{FN} ({SF})' e.g. "0.88 (0.88)", and push either qwen base or think to appendix as they mostly match.
- Increase figure font size to match text, and remove the graph titles (put necessary information in the caption).
- "MPC" -> "MCP" page 7 paragraph 3.
- Top of page 12 is referencing sections in appendix.
- Slot filling metric may be less of an issue with the features like the outlines library [1] which can enforce correct output structure at the token selection stage of generation - if function name is called, the parameters can be assumed to be valid.

[1] https://github.com/dottxt-ai/outlines

**Audience:**

Yes

**Audience Explanation:**

This new problem setting is very important as function calling models become more widely used. The threat vector of malicious tool descriptions is realistic as tools (functions) may not be carefully checked and they may avoid the malicious prompt checks done on the incoming user prompts.

**Broader Impact Concerns:**

Broader impact is sufficiently addressed.

**Claims And Evidence:**

No

**Claims Explanation:**

Abstract/intro claims imply a single attack is robust to payload and query perturbations, but experiments only support robustness to individual setting changes. The experiment variants also require different optimizations, which makes it difficult to say which single configuration is the primary proposed FHA attack.

In the attack setting proposed, an attacker wouldn't necessarily have much information about the target context: functions (and their order),  user query, or even the model being used are unknowns. The core evidence for effectiveness should show the attack is effective despite these unknowns, but primary results (Table 2) shows a white-box setting where the full model and context are known.

Jailbreak ASR is typically formulated as a success if at least one attempt is successful. However, changing the goal to getting a model to make a function call makes the "at least one" not suitable. This impacts GCG's early stopping, as GCG assumes ASR "at least one" is the target metric, potentially inflating Table 2 results. Removing the max from ASR makes the metric match the more realistic setting of the LLM not being used with greedy decoding.

Simple defenses like perplexity checks can effectively filter text optimized with GCG, which may go against the claim of the need for strong safety modules.

**Requested Changes:**

(critical) **Improve clarity of the method**. Section 4 should formulate the optimization of FHA and not just describe the changes to Eq. 6,7 in the following paragraphs. Additionally, it would improve experiment clarity if the optimization changes of the "universal" settings were noted in Section 4. Adding names to the different settings (already loosely named in headings like "formulation diversity") would help clarify what kinds of robustness are being evaluated. This belongs in section 4 because it's a part of the universal FHA method being proposed, though an instance where all options are added would simplify presentation (individual optimizations become ablations).

(critical) **Improve evidence for robustness.** Transferability of the attack should be evaluated under a more realistic black-box setting, with function order, other functions, query, and target LLM all unknown. This would help label a single method as the proposed FHA method - otherwise the proposed contribution may reduce to evaluating GCG with different optimizations in the function calling setting. Alternatively, weakening claims made in abstract/intro to reflect the individual settings evaluated would resolve this (albeit with less contribution).

(semi-critical) **Improving metric.** Don't use greedy decoding for reported ASR - changing criteria from max to an average success. Getting multiple responses per sample would make (average) ASR more accurate, but is likely not required with a dataset of 200 samples.

(not-critical) **Evaluation on frontier model.** Evaluating on frontier models would help show the vulnerability of the current strongest models and would show whether the risk is limited to smaller models. The motivation for evaluating on smaller target models in Section 5 makes this not critical and would just strengthen the paper's results.

(not-critical) Figure 6 is missing 3 ->1,2. Important for comparing the impact of optimizing multiple intents on more narrow domains.

(semi-critical) Figure 5 would be more informative if transfer were evaluated across entire dataset. Can take chosen suffixes and directly transfer them rather than testing transfer ASR during optimization.

---

> ### Author Response · Authors · 2026-07-04
> **Answer to Reviewer**
>
> Thank you for your review comments and feedback on our work. We appreciate your questions and your concerns about our work. It helps us to better understand our weaknesses and gives us ways of enhancing and strengthen our claims. To address your concerns, we uploaded an updated version of the manuscript, with changes annotated in blue to better highlight our modifications. Please find below our responses to your questions.
>
> ### 1. Improve clarity of the method (critical)
>
> We agree that Section 4 under-motivates our threat model with regards to the existing literature, and currently describes the base FHA objective and only describes the universal variant in Section 7. In the revised version, we included the following changes:
>
> - Reformulated Section 4 to state the universal optimization objective (Eq. 8 - Eq. 6–7 plus the batch-accumulated loss from Algorithm 2) explicitly, rather than deferring it to Section 7.
> - Named each robustness setting consistently ("Formulation diversity," "Argument variation," "Multiple intents," "Batch of position," "Batch of number") and introduced them in Section 4.
>
>
> ### 2. Improve evidence for robustness (critical)
> We included an ablation transferring the attacks from one model to another (see Table 5 in Section 7). We found the attacks transfers modestly to other model families or sizes, but that the ASR obtained is non-negligeable (11.2-27.6% ASR). This is likely due to the formatting of the optimised suffixes. For this reason, we still report these findings for transparency, but do not claim full cross-model transferability of our attack.
>
> ### 3. Improving metric (critical)
> While greedy decoding appears to be a standard to test attacks in the literature, we agree that evaluating the average ASR across runs would be useful. To address this concern, we included a comparison between max and average success, by inferring models 10 times per query and report the standard and standard deviation per samples. Table 7 in Appendix L presents this experiment. Importantly, the average ASR closely match the max ASR score of FHA across models.
>
> ### 4. Evaluation on frontier model (not-critical)
> We agree that evaluation of FHA on Frontier models would be valuable, but such study is out-of-scope of our paper. We mentioned in the Target models paragraph in Section 5 that FHA is primarily a white-box attack as it requires gradient access to the target model, and most Frontier models are currently close-source. While transferability to these models might be feasible, we leave that to future work, and only present transferability score on open-source models.

---

> > ### Comment · Reviewer_hzRk · 2026-07-09
> >
> > Thank you for the detailed response and revisions. My concerns have partially been addressed.
> >
> > 1. **Improved clarity.** Equation 8 and named robustness settings help clarity. In section 4, when describing "Formulation diversity" and "Argument variation", it would help to use the $\\hat{y}\_{fh}$ and $a\_{target,i}$ notation defined in equation 5 to clarify how the query is changing.
> >
> > 2. **Improved evidence:** The model-transfer results are very interesting. The wording can be more positive though: that the attack *can* be transferred between models, rather than them being "non-negligeable". It's difficult to judge whether the numbers are high or low due to lack of comparable methods. A best-of-N baseline similar to [2], where a suffix is randomly generated 100 times and ASR (max or average) is reported, may show the numbers observed are significantly better than that baseline. This is not critical, though it would help contextualize the results.
> >
> > 3. **Improved metric:** A 6-14% drop is significant. Greedy decoding is less of an issue when the evaluation data is not being optimized on - my understanding is that this is the case for MPMA. Because GCG directly optimizes over the evaluation prompts, the comparison in Table 2 may be unfair and its ASR inflated by measuring success on the same contexts used for optimization. This is less of an issue when evaluating a jailbreak, where the threat model only requires on harmful completion, but is important for the new proposed setting, where reliable and consistent success is the relevant objective.
> >
> >
> > Typos:
> > - intro, bullet points at end, first bullet point, last sentence "This FHA’s primary setting"
> >     - 2nd bullet point "evalauted"
> >
> > [2] https://github.com/jplhughes/bon-jailbreaking

---

> > > ### Author Response · Authors · 2026-07-11
> > >
> > > Thank you for your follow-up answer. Please find below answers to your questions:
> > >
> > > 1. **Improved clarity.**
> > > We agree that the notation of the target and its parameters was missing, and helps clarifying the differences across batches. We suggest a revised version of Section 4, where we discussed about this aspect. It is worth noting the difference between $y_{ground truth}$ (the tool-call that is generated by the prompt without attack), and $\hat{y}_{target}$ (the tool-call that the attack seeks to trigger). We precise in our revision that:
> > > - For Formulation diversity, both names and arguments of $y_{ground truth}$ are identical across queries, and $\hat{y}_{target}$ is the same.
> > > - For Argument variation, the names of $y_{ground truth}$  are identical across queries, but each arguments $a_{target, i}$ changes. In this case, $\hat{y}_{target}$ is the same because we only includes the name of the target in the objective.
> > >
> > > 2. **Improved evidence.**
> > > We agree that the wording can be enhanced. We applied modification to be more positive about our cross-model transferability findings. Due to time constraints, we were limited in the computations that we could perform. First, to address this concern, we suggest to adopt a similar setting as per Table 7 in Appendix L (non-greedy decoding, average ASR across 10 runs per queries). We believe that such metric can enhance our cross-model evaluation. Second, we agree that adding baselines could be relevant here. BON [2] is a nice suggestion. We also suggest to add the cross-model transferability scores of our novel baseline, the multi-round function injection. This could be a fair baseline since this new baseline shares the same attack setting as FHA.
> > >
> > > 3. **Improved metric.**
> > > We agree that the drop is significant in Table 7 (Appendix L). However, we justify that this is due to our experimental settings. For reproducibility, we deliberately set the inference to greedy decoding, and used seeds (set to 42). This setting was used for both training (FHA's optimization), as well as for evaluation. Table 7 (Appendix L) presents the results of optimization conducted using greedy decoding (seed 42), but the evaluation using non-greedy decoding (sampling). We believe that this setting under-estimate the performance of the algorithm. Indeed, we suspect that optimization using non-greedy decoding can result in enhanced performance of FHA while evaluating it.
> > >
> > > Importantly, other baselines in Table 2 are also using the prompt when optimizing the attack, which makes the evaluation fair because other baselines has the same context information as FHA:
> > > - Function Injection (ZS) takes the input prompt
> > > - Function Injection (FS) and MPMA take the input prompt as well as all existing functions of the payload
> > >
> > > 4. **Typos.** We apologies for the typos and addressed the issues that were pointed in our revised version.